# A high-density crowd state judgment model based on entropy theory

**Guomin Zhao, Cong Li, Guangji Xu, Falong He\*, Jing Zhang**

School of Energy and Safety Engineering, Tianjin Chengjian University, Tianjin, PR China

\* hiroki_ho@foxmail.com

## Abstract

A high-density crowd state is prone to cause large-scale crowd stampede accidents that seriously threaten the people's security and property. The key to preventing crowd congestion is to accurately predict the location and time of crowd events, particularly when there is a high density of people. In this paper, the entropy theory is used to characterize the state of a crowded system. The theoretical entropy $S_r$ and the actual entropy $S$ of the crowd system are obtained according to the area occupied by the different crowd state The maximum entropy value and the actual entropy value under different conditions of the system are compared to judge the state of crowded extent. The results show that the model is practical and effective. According to the situation of the crowd, different management and evacuation measures are considered to prevent the occurrence of crowd accidents.

## Introduction

With the rapid development of China's economy, the number of large public gathering activities has increased, and this creates an increased amount of accidents in crowds that can result in serious casualties. The occurrence of sudden accidents can easily lead to a massive loss of life and property. Therefore, a reasonable judgment of crowd status is of great significance to prevent such social and public safety accidents.

Under the emergency, the crowd state will appear abrupt change because of the unexpected release of people pressure. The accidental release of energy causes the group to shift suddenly. The individual among the crowd will lose self-control and even result in a fall or trample accident [1–2]. At present, scholars mainly judge the abnormal state of the crowd based on image processing [3–5]. However, the image processing method is difficult to predict the behavior state of the crowd, especially for poor lighting conditions. The system entropy reflects the degree of chaos to a certain extent [6]. Singh et al. [7] applied entropy theory to behavior prediction and management decision support, and used entropy value to detect abnormal crowd behavior. Saira [8] estimated the crowd social entropy and used it to describe the existence of uncertain behaviors. Zhao [9] established an entropy model to describe the macro state of group behavior and explored the change of entropy caused by the mutation of the group behavior. Huang [10] defined speed entropy as the basis of congestion detection. The results

**Data Availability Statement:** All relevant data are within the manuscript and its Supporting information files.

**Funding:** This work was supported by the National Natural Science Foundation of China under grant

51108297 and Research Project of Tianjin Education Commission under grant 2018KJ168 and 2020KJ043.

**Competing interests:** The authors have declared that no competing interests exist.

show that the method can detect abnormal crowd behavior in real time without the need to identify and track pedestrians.

It is worth noting that previous studies mainly focused their attention on the behavior and the entropy value of low-density populations, while less attention was paid to the state evolution of the high-density populations. In fact, the state evolution of the high-density population is the critical factor for the gathering, crowding and trampling accidents. In this paper, the population is divided into four states: free flow, maximum flow, intermittent flow and crowed state of confusions. By calculating the entropy value of each crowd state, a judgment model of crowd state expressed by local entropy is established. The model could clearly describe the entropy change of the crowd density from high to low, and predict whether the population density or capacity in a specific area is at its maximum or not. A reliable value can also be generated in a specific area of a crowd accident that will allow the appropriate measures to be taken to reduce casualties.

## High density population status analysis

The stationary and moving population's critical safety density is 4.7 pedestrians/m$^2$ (4.7 ped/m$^2$) [11], respectively. It has been discovered, when the crowd density reaches 5 ped/m$^2$, the velocity of a person will be obviously influenced. With a further increase in crowd density, congestion will occur. The pedestrian flow in this stage is called the crowding flow, and the crowd density is too high. When the crowd density is high, the distance between individuals will be reduced, which will cause pushing among the individuals. Thus, the group will not be completely at rest and move at a certain speed. The individuals will possibly fall but have no chance to stand up, which will result in a stampede. The crowd density tolerance of different regions varies [12], as shown in Table 1. With an increase in the crowd density, the feelings of an individual will change [13], as shown in Table 2. The study found that if the crowd density is 6–8 ped/m$^2$ for a long time, lung oppression and contraction will occur, which will result in breathing difficulties, heart and brain hypoxia, and other issues. This may even lead to crowd casualties [14]. When the average crowd density was 6 ped/m$^2$, the local crowd density could reach 9 ped/m$^2$, but a crowd with extreme density keeps moving (the average velocity is 18 m/min). The pedestrians gathering behavior could be observed when the local crowd density was greater than 6 ped/m$^2$. For this state, the individuals will lose self-control and act as mobile substances in the crowd flow, causing pushing among the pedestrians. Even the forward or backward pressure wave will be observed in the crowd.

According to the crowd density, state characteristics, and findings of this study, the population status was divided into four cases: free flow, maximum flow, intermittent flow, and crowed state of confusions. Suppose the pedestrian density is in a very crowded state, the individual experiences a sharp space reduction. The space of the individual will be irregular due to the surrounding push, which will most likely cause packed crowds.

The relationship between the population system and entropy theory can be drawn according to the characteristics and individual status analysis. It also introduces crowd pressure as a

**Table 1. The average maximum tolerable density of a population.**

| Countries and regions | Sex | Maximum tolerable density (ped/m$^2$) | Countries and regions | Sex | Maximum tolerable density (ped/m$^2$) |
|---|---|---|---|---|---|
| United Kingdom | Male | 8.76 | Hong Kong | Male | 11.5 |
| United Kingdom | Female | 9.92 | Hong Kong | Female | 10.8 |
| Western Europe | Male | 7.42 | Japan | Male | 11.7 |
| Western Europe | Female | 8.18 | Japan | Female | 12.8 |

**Table 2. Relationship between crowd density and state.**

| Crowd density (ped/m$^2$) | Standing state (1m$^2$) | Crowd density(ped/m$^2$) | Standing state (1m$^2$) |
|---|---|---|---|
| 5 | Contact with the next person's clothes | 8 | Between people and people can barely squeeze |
| 6 | Can pick up goods beside the foot | 9 | Hands cannot move up and down |
| 7 | Shoulder and elbow feel pressure | 10 | Feels stressed, unable to move, and calls for help |

generalized force and crowd flow as a generalized flow. The product of generalized force and generalized flow is used to determine the local entropy generation. Then the entropy value model is constructed to judge the different states of the crowd. A flowchart for creating the entropy judgment model is shown in Fig 1.

## High-density crowd system modeling based on entropy theory

### Entropy theory

Entropy is the order of a thermodynamics system. The larger the entropy, the more disordered the system. The smaller the entropy, the more ordered the system. For an open system, the total entropy is $dS = d_iS + d_eS$, where $d_iS$ is the system-generated entropy change, which is greater than or equal to 0; and $d_eS$ is the positive and negative entropy changes, which flow from the outside into the system. The energy and material exchange of the system and the outside world will cause entropy flow. Negative entropy flow obtained from the outside can reduce the total entropy of the system, make the system develop in an orderly direction, and keep the system in a stable state.

### Crowd system entropy and the entropy model

Individual behavior is significantly influenced by the surroundings in a population system. If the crowd density is relatively low, the crowd flows freely. It does not have the essential characteristics of a continuous medium. Because a population has self-organizing characteristics, a population quickly returns to an ordered state after a disturbance. Thus, self-organization is regarded as negative entropy flow, i.e., $dS = d_eS \leq 0$, where the total entropy of the system is reduced, stable, and in an orderly state.

When the density is high, a crowd can be regarded as a continuous medium, and any disturbance in the crowd will be transmitted in the form of a crowd wave. At the same time, differences between individual groups will result in a nonlinear distortion of the wave. Thus, an individual loses the ability for self-control. As a result, a stable and orderly state cannot be restored in the crowd through self-organization, which means that $dS = d_iS \geq 0$. It is difficult to move in a crowd out of control, which would lead to suffocation or even a crowd stampede and casualties.

**(1) Local entropy production.** Considering the relationship between crowd characteristics and entropy theory, the local crowd entropy can be described by the generalized force and the generalized flow. Generally, the force is the cause of flow. The greater the generalized force, the greater the flow, the greater the entropy and the greater the disorder within the system. Local entropy production can be written as follows:

$$\delta = JX \tag{1}$$

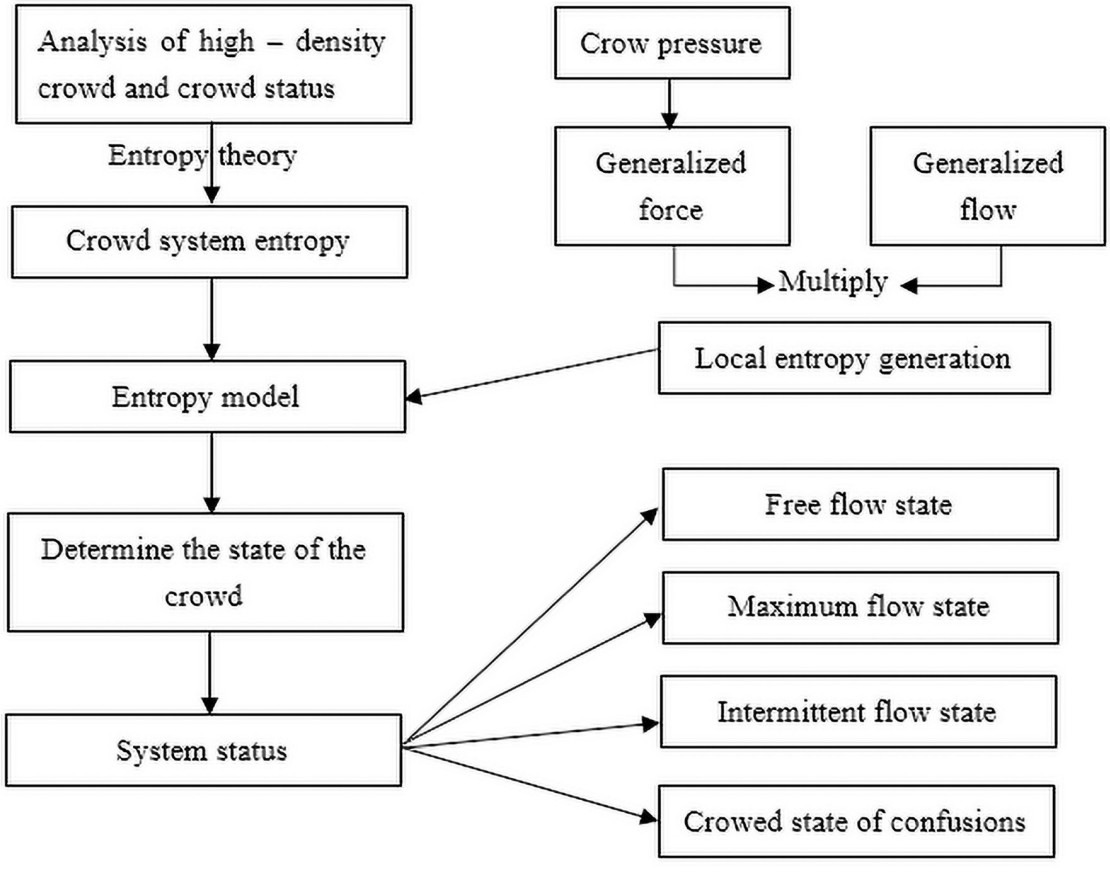

**Fig 1. A flowchart for establishing the entropy judgment model.**

where $\delta$ is the local entropy production in the system, X is the generalized force, refers to crowd pressure and J is the generalized flow, refers to the number of people.

For an open system, the nonlinear interaction between generalized force and flow is the main factor for a crowd to reach a stable and orderly state.

**(2) Generalized forces in a crowd.** Emergent events are caused by a sudden release of pressure due to a change in the local force size and direction of the force chain formed in the crowd. The pressure distribution in a crowd, as shown in Fig 2 [15], can be expressed using the product of the local crowd density and the change in local flow velocity:

$$\vec{P} \;=\; \rho(\vec{r})\Delta v(\vec{r}, t), \tag{2}$$

where $\vec{P}$ (ped/(m*s)) represents the generated crowd linear kinematic pressure, $\rho$ (ped/m$^2$) represents the linear crowd density (note that this is the square root of the area density), $t$ (s) represents the movement time of the crowd, $\vec{r}$ (rad) represents the angle between the velocity and the target region, and $\Delta v$ (m/s) represents the change in the velocity of the crowd that drives the pressure variation.

The current research system is a specific area where people can pass in and out freely. There is a phenomenon of people gathering at a specific position in this area, but it is no stampede accident. It is noted that there is no allowance made here for the coefficient of stress relaxation commonly associated with shear stress between the feet of those persons in the crowd and the

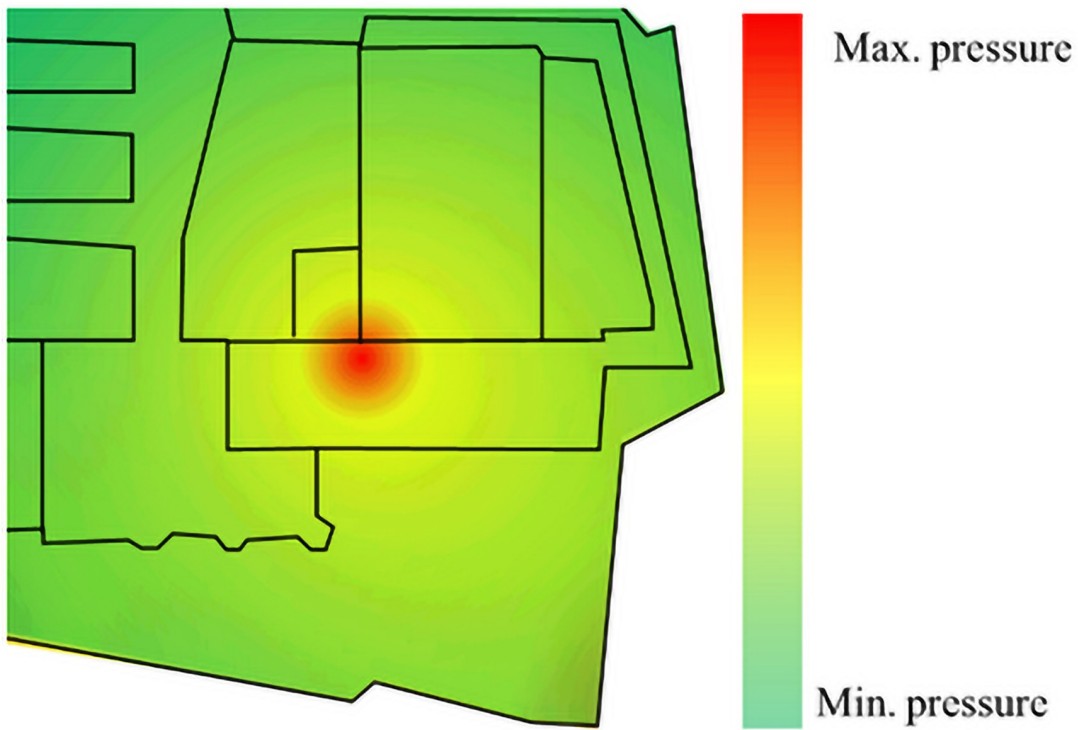

**Fig 2. Simplified diagram of crowd pressure distribution.** Green means minimum pressure, yellow means medium pressure, and red means maximum pressure. When the colors are the same, the darker the color, the greater the pressure.

underlying ground. This coefficient represents the inverse of an individual pedestrian's internal adjustment time and has units of 1/s. The time scale is presumed small compared to those of the adjustments considered here. However, because the internal adjustment time is considered constant, this dimensional coefficient is used to normalize the pressure partially.

**(3) The establishment of local entropy production.** The entropy production rate of a crowd system indicates the velocity of the change in the system state. The group pressure is regarded as the generalized force in a crowd, and the individual pressure in different regions of a crowd is simplified, as shown in Fig 3. The green area is free flow, the red area is maximum flow, and the purple area is intermittent flow. As mentioned above, the group pressure leads to the production of crowd entropy. Thus, the local entropy crowd in the system can be expressed as

$$\delta \; = \; \Delta(\vec{P}N) \tag{3}$$

where $N$ (ped) is the number of people in the study area, $N \; = \; \rho(\vec{r})\Delta A$: $\rho$ (ped/m$^2$) is the regional crowd density;$\Delta A$ (m$^2$) is the study area.

The entropy production of the small region in the system is as follows:

$$\delta \; = \; \Delta(PN) \; = \; \rho(\vec{r})\Delta v(\vec{r}, t)\rho(\vec{r})\Delta A \; = \; \rho^2(\vec{r})\Delta v(\vec{r}, t)\Delta A \tag{4}$$

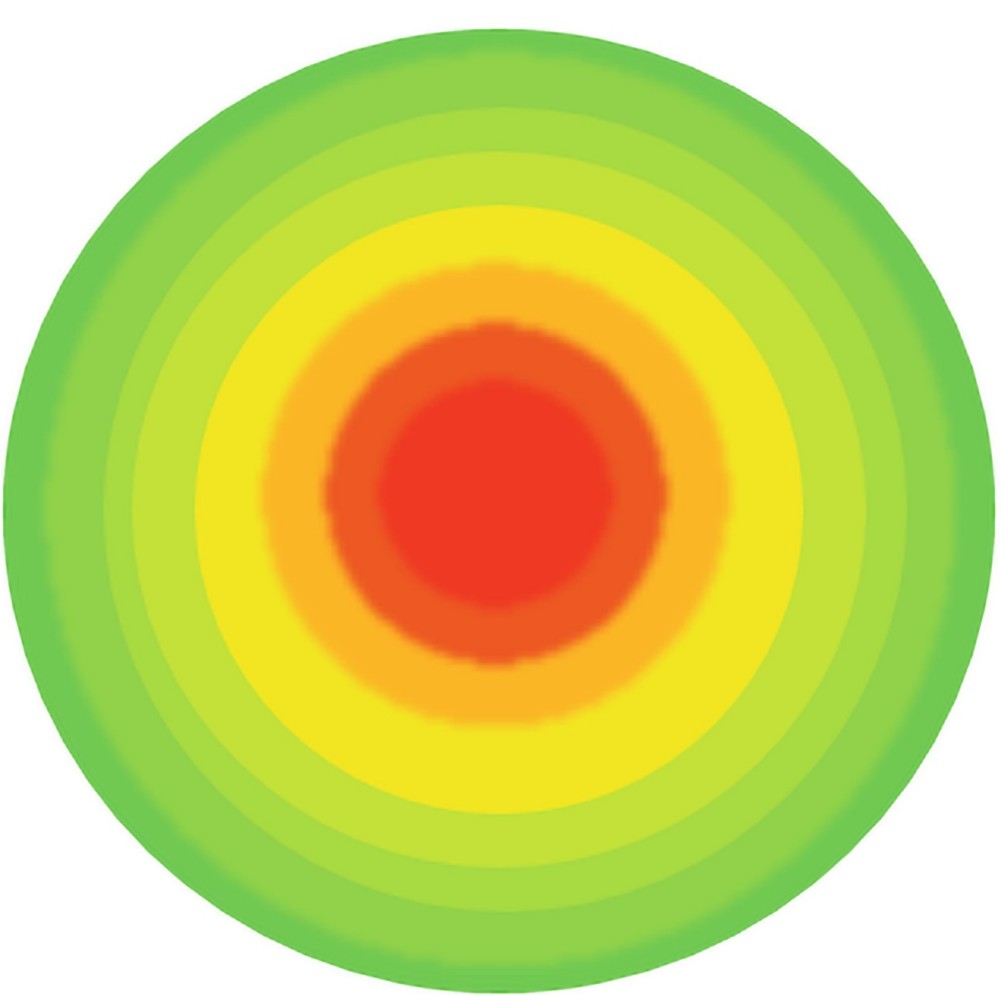

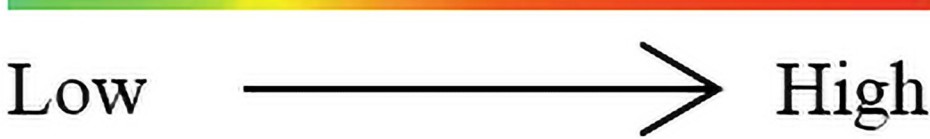

**Fig 3. Simplified diagram of crowd pressure.**

According to the local equilibrium assumption, the studied area entropy production is as follows:

$$p_i = \int_A \delta dA = \int_A PN dA = \int_A \rho^2(\vec{r})\Delta v(\vec{r}, t) dA \tag{5}$$

According to Eq (4), the entropy production is the function of the crowd density, velocity change, and studied region area.

Entropy is a state variable that is irrelevant to the system process. When the system reaches the equilibrium state, entropy is completely determined and irrelevant to the path to equilibrium. Entropy is also an important parameter to describe the degree of system chaos. Generally, the smaller the entropy, the more orderly the system can reach. When the system entropy $S_0 = 0$, the initial velocity will be constant. Thus, the velocity variation, $\Delta v = 0$, and the personnel flow pressure $P_0 = 0$. The entropy generation model is improved and expressed as follows:

$$p_i \;=\; S - S_0 \;=\; \int_A \int_{\rho_0}^{\rho} \rho^2(\vec{r}) \Delta v(\vec{r}, t) d\rho dA \tag{6}$$

## The crowd state judgment model

### Model description

In the initial state, the system entropy values, velocity variation, and crowd pressure are 0. In this case, the crowd is in free flow. Thus, the entropy of any other equilibrium can be expressed as follows:

$$S \;=\; \int_A \int_{\rho_0}^{\rho} \rho^2(\vec{r}) \Delta v(\vec{r}, t) d\rho dA \tag{7}$$

The system entropy and entropy generation are different for the steady and unsteady crowd system.

**(1) Steady crowd system.** In the steady phase, the velocity and density of a crowd do not change with time. The inflow and outflow of people in the studied area are the same, and the entire area is in an equilibrium state. The entropy production steady stage is $p_i = 0$, and the entropy can be expressed as follows:

$$S \;=\; \int_A \frac{1}{3}\left(\rho^3 - \rho_0^3\right) \Delta v(\vec{r}, t) dA \tag{8}$$

**(2) Unsteady crowd system.** Under the condition of high crowd density, some people will impact the normal crowd because of a sudden acceleration or emergency. It is assumed that in a short time, the crowd velocity changes from $v_1$ to $v_2$, and the density changes from $\rho_1$ to $\rho_2$. The system entropy generation is expressed as follows:

$$p_i \;=\; S - S_0 \;=\; \int_A \int_{v_1}^{v_2} \int_{\rho_1}^{\rho_2} \rho^2 \Delta v(\vec{r}, t) d\rho dv dA \tag{9}$$

### The crowd state judging model

When the flow of people reaches the maximum value, the group and individual velocity decrease sharply with increased crowd density. When the flow of people is the highest, the entropy of the crowd is expressed by $S_1$. When the crowd density increases to a specific value, the crowd cannot move smoothly, and the crowd can only move intermittently. For this situation, the entropy of the crowd is expressed by $S_2$. When the system entropy is greater than $S_2$, crowd congestions or confusions occur.

**(1) Determining system entropy.** In the initial state of a crowd, the maximum velocity is $v_0$, the crowd density is $\rho_0$, the rate of entropy generation is 0, the crowd pressure is 0, and the system is in equilibrium. Thus, the system entropy is 0

$$S_0 \;=\; 0. \tag{10}$$

Research results show that when the flow velocity is $1/2v_0$, the flow of people reaches the maximum, and the crowd density is $\rho_1$. Thus, the system entropy can be expressed as follows:

$$S_1 \;=\; \int_A \frac{1}{3}\left(\rho_1^3 - \rho_0^3\right)(v_0 - v_1)dA + S_0 \;=\; \int_A \frac{1}{3}\left(\rho_1^3 - \rho_0^3\right)(v_0 - v_1)dA \;. \tag{11}$$

When the crowd begins to jam, the crowd density is $\rho_2$, and the crowd velocity is $v_2$. At this time, the crowd system entropy can be expressed as follows:

$$S_2 \;=\; \int_A \frac{1}{3}\left(\rho_2^3 - \rho_0^3\right)(v_0 - v_2)dA + S_1. \tag{12}$$

**(2) Determining the state of the crowd.** In this study, the state of a crowd is divided into four: types free flow ($S = S_0 = 0$), maximum flow ($S_0 < S \leq S_1$), intermittent flow ($S_0 < S \leq S_2$), and crowed state of confusions ($S > S_2$).

## Definition of crowd system state

Crowd speed and density are the keys to determine the entropy value of the system. Scholars have studied the relationship between the velocity and the density of a crowd. Fig 4 shows a summary of the relationship between the moving velocity and the density of a crowd [16]. All model data is provided in S1 Data.

The crowd velocity decreases rapidly with the crowd density increases, as Fig 4 shown. Due to the difference in researches, such as observation sites and methods, data analysis and processing methods, sample size, there are differences among the models. Except for P&M model, Ando model and Zhang Peihong model, most of the models are within the range of crowd density with $\rho \leq 4$ ped/m², and the results are relatively close. The movement speed and crowd density show a linear relationship, which conforms to crowd safe flow state characteristics. However, when the crowd density is greater than 5 ped/m², the crowd velocity is not equal to 0. This indicates that the crowd did not stop moving. The Green Guide studies the relationship between average population speed and crowd density [17, 18]. Combined with the Ando model, this model studies the crowd density and speed in a highly dense population. The calculated crowd flow is shown in Table 3.

As can be seen from Table 3, in the specified area, the higher the speed, the smaller the crowd density, and vice versa. When the crowd density is less than 1.82 ped/m², the crowd velocity is greater than 1 m/s, which is the free-flow state. When the crowd velocity of is approximately half of the maximum velocity, i.e., 0.6 m/s, the flow state is called the maximum flow state. When the velocity of the crowd is approximately 0.25 m/s, the flow state is called intermittent flow. With further increase of the crowd density, the crowded flow state would reach the confusion state. According to Fig 4 and Table 3, the crowd status definition criteria are obtained as shown in Table 4.

In order to prevent the crush and stampede accident, it is necessary to carry out early warning and control measures. A timely and accurate warning or evacuation guide before an accident occurs is important to reduce the crowd pressure, the total system entropy and the crowd disorder degree, and protect the personnel's life and safety.

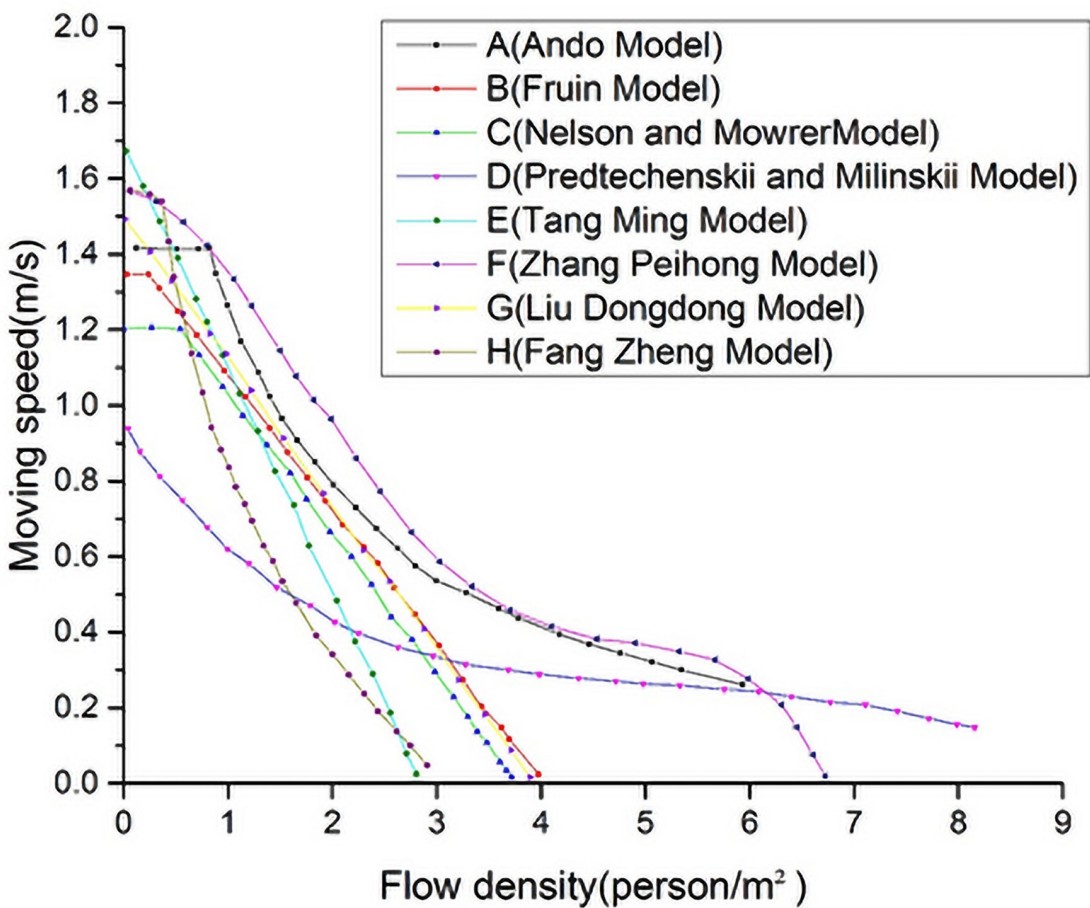

**Fig 4. Summary of the relationship between the moving velocity and the density of a crowd.**

## Model validation study

In order to verify the performance of the established model, this part builds a simulation scene based on AnyLogic (AnyLogic 8.7.4 Personal Learning Edition). Then the validity of the population status judgment model is verified.

**Table 3. Data flow calculated using the Green Guide.**

| Average velocity (m/s) | Average density (ped/m$^2$) | Flow (ped/(m$^*$s)) | Average density (ped/m$^2$) | Average velocity (m/s) | Flow (ped/(m$^*$s)) |
|---|---|---|---|---|---|
| 1.30 | 1.40 | 1.82 | 0.5 | 1.34 | 0.67 |
| 1.20 | 1.51 | 1.812 | 1.0 | 1.34 | 1.34 |
| 1.10 | 1.65 | 1.815 | 1.5 | 1.21 | 1.815 |
| 1.0 | 1.82 | 1.82 | 2.0 | 0.91 | 1.82 |
| 0.9 | 2.02 | 1.818 | 2.5 | 0.73 | 1.825 |
| 0.8 | 2.27 | 1.816 | 3.0 | 0.61 | 1.83 |
| 0.7 | 2.60 | 1.82 | 3.5 | 0.52 | 1.82 |
| 0.6 | 3.03 | 1.818 | 4.0 | 0.45 | 1.8 |
| 0.5 | 3.63 | 1.815 | 4.5 | 0.40 | 1.8 |
| 0.4 | 4.54 | 1.816 | 5.0 | 0.36 | 1.8 |

**Table 4. The crowd status definition criteria.**

| crowd status | Crowd velocity (m/s) | Crowd density (ped/m$^2$) | Entropy |
|---|---|---|---|
| free flow state | $\geq 1.0$ | $\leq 1.82$ | $S_0$ |
| the maximum flow state | 0.6–1.0 | 1.82–3.0 | $S_1$ |
| intermittent flow | 0.25–0.6 | 3.0–5.6 | $S_2$ |
| crowed state of confusions | $\leq 0.25$ | $\geq 5.6$ | $S > S_2$ |

The social force model is used to model pedestrian movement in AnyLogic. We designed a illustrative experiment on pedestrian congestion in a straight passage, shown as Fig 5. In the simulation case, people move from two sides of the passage to the other side, and 500 pedestrians appear at each time step. Over time, the pedestrians in the simulation become denser until the congestion occoured. All model data is provided in S2 Data.

Through the statistics of pedestrian speed, the crowd system entropy value is calculated according to Eqs (10)–(12). As can be seen from Fig 6, at the beginning of the simulation, pedestrians move freely and their speed is stable at about 1 m/s. As time changes, pedestrian speed basically decreases in a linear relationship and the entropy of crowd system gradually increases from zero in the initial state. When the time step is 320 to 360, the speed value stabilizes at about 0.65m/s. After the time step 360, the speed begins to decrease gradually, and the entropy increases with the decrease of the speed. This is because that the distance between pedestrians have no influence to their speed scine the adequate spacing at the initial state. With the increase of the number of pedestrians, the repulsive force between pedestrians become dominat during the movement, which would result in a the decrease of pedestrian speed and the chaos of the pedestrian movement state. The congestion state arising from the pedestrians crowding together, affected the moving speed of pedestrians and increased the crowd system entropy value. According to the simulation, the entropy of a system is related not only to population density, but also to the actual area of the system occupied by people of different densities. That is to say that the larger the system area occupied by high density population, the greater the crowd system entropy.

## Case analysis

In this chapter, we calculate crowd system entropy according to the crowd status judgment model to determine the crowd state and then combine it with the early warning system to warn the occurrence of crowd crowding events. In practice, the warning level can be established based on the judgment results, and different management measures can be taken for different warning levels. From high to low warning levels are divided into red alert, orange alert, yellow alert.

The "yellow alert" indicates that the population is beginning to experience abnormalities. The "orange alert" means that the population is on the alert, and it needs to pay timely

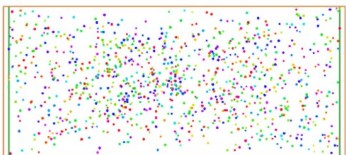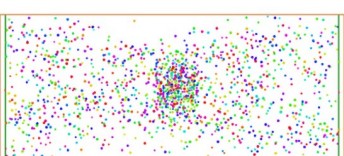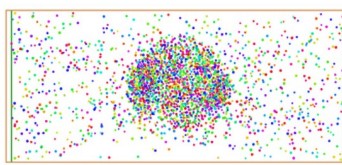

**Fig 5. The state of pedestrian movement in the simulation.** The brown line represents the boundary, the green line represents the pedestrian target, and the colored dot represents the pedestrian.

attention to and adjust and control relevant key influencing factors. The "red alert" means that the crowd is at a high risk of crowding state. The managers should immediately implement the emergency treatment plan to avoid the crowd crowding event causing significant loss of life and property.

According to the crowd status judgment model, to confirm the entropy of the crowd system with different densities, the area occupied by people of different states in the entire system must be determined first. According to the size of the research system, different research radius, R, can be defined. In this study, an assumption is that the radius of the studied area, R, is 5 m. The video monitoring can estimate the number and density of people in a definite region. Then, the maximum entropy value under different conditions of the study system can be obtained. Finally, the actual value and the maximum value can be compared to judge the system state. It is assumed that the crowd density does not change over small time intervals and only the velocity changes, and the crowd density has a linear distribution in the studied area. Generally, the highest crowd density is at the center of a certain population circle.

(1) When the observed crowd density is $\rho < 1.82$ ped/m$^2$, the system entropy can be expressed as follows,

$$S_0 = 0$$

The crowd is in free flow state, and there will be no crush and stampede accident, so no control measures are needed.

(2) When the observed crowd density is 1.82 ped/m$^2 < \rho \leq 3$ ped/m$^2$, R$_1$ is the radius of the theoretical area of the maximum flow state. $R_1 = \frac{5}{3}(3 - 1.82) = 2$m, $A_{1,r}$ represents the theoretical area of the region occupied by the population in this state. With $\rho_1 = 3$ ped/m$^2$, $\rho_1 = 1.82$ ped/m$^2$, $v_0 = 1$m/s, $v_1 = 0.6$m/s, substituted into Eq (11), $S_{1,r}$, represents the upper limit of theoretical entropy of the maximum flow in the region, $S_{1,r} = 35$. When the actual area of the system occupied by the maximum flow group is less than 12.56m$^2$, the entropy value of the system is

$$S < S_{1,r}.$$

In this state, the number of people increases gradually, but it has not reached the maximum value. Therefore, activating the yellow alert and measures are strengthened to prevent the crowd state from turning into the intermittent flow state, such as strengthening the warning function of evacuation signs.

(3) When the observed crowd density is 3 ped/m$^2 < \rho \leq 5.6$ ped/m$^2$, R$_2$ is the radius of the system theoretical area occupied by the intermittent flow state. $R_2 = \frac{5}{5.6}(5.6 - 3) = 2.3$m. R$_1$ is the radius of the theoretical area of the maximum flow state. $R_1 = \frac{5}{5.6}(5.6 - 1.82) = 3.3$m. With $\rho_2 = 5.6$ ped/m$^2$, $\rho_0 = 1.82$ ped/m$^2$, $v_0 = 1$m/s, $v_2 = 0.25$m/s, substituted into Eqs (11) and (12), $S_{2,r}$ represents the upper limit of theoretical entropy of the maximum flow in the region. $S_{2,r} = 2.8\pi R_1^2 - R_2^2 + 42.4\pi R_2^2 = 746.4$. When the actual entropy of the system is

$$S < S_{2,r},$$

crowded situation increases and congestion is imminent. In order to prevent the further deterioration of the situation resulting in accidents, the early warning system launched an

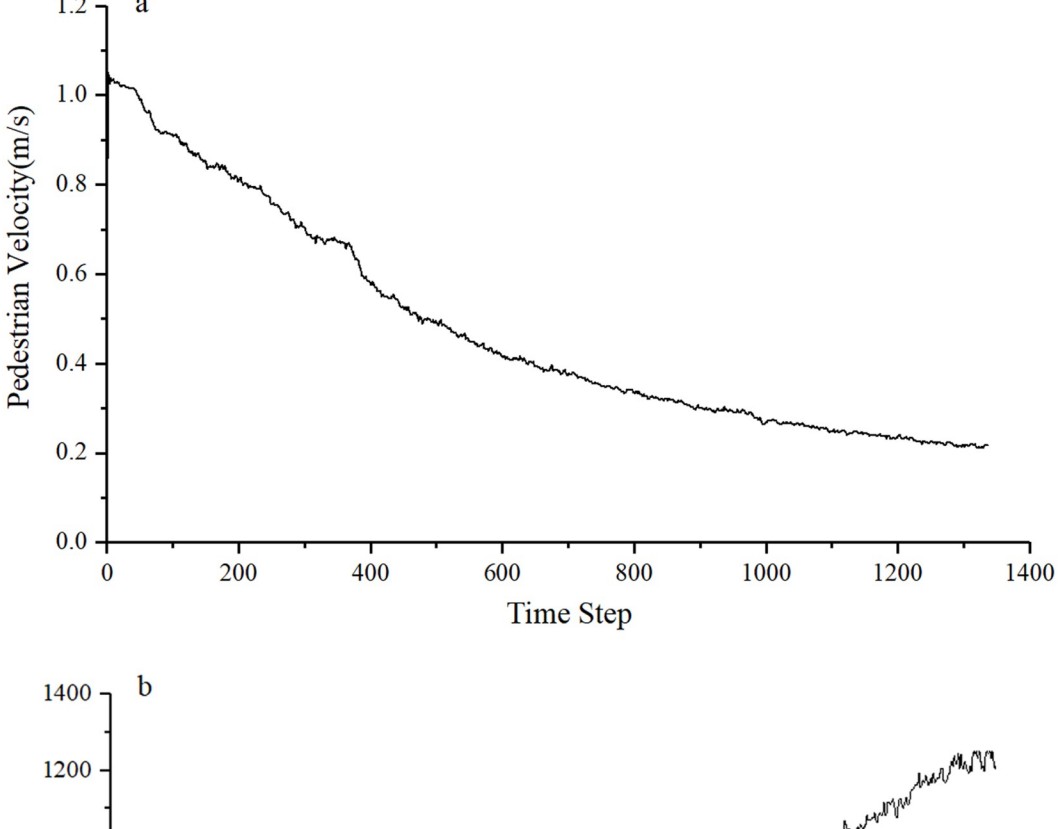

**Fig 6. Simulation results of pedestrians walking in a straight passage with the size of 5m×25m(length × width).** (a) Pedestrian movement velocity. (b) Crowd system entropy.

orange alert. Security personnel should be increased on the basis of the original, to reduce crowd gatheringand prevent the local crowd density too large.

(4) When the observed crowd density is $\rho > 5.63$ ped/m$^2$, the entropy value of the system is

$$S > S_{2,r.}$$

The crowd state has become crowed state of confusions. At this situation, the red alert should be started. The crowd should be evacuated immediately. The compulsory measures should be adopted for the crowd dredging and diversion, to prohibit the influx of people.

## Discussion

In the present work, an entropy model is established for the high density population system. The entropy value can be used to judge the crowd state in the area at a certain time in the future. The crowd system state is divided into the free flow ($S = 0$), maximum flow ($S \leq 35$), intermittent flow ($35 < S \leq 746.4$), and crowed state of confusions ($S > 746.4$). The results show that the crowd system entropy is not only related to the population density, but also related to the actual area occupied by people with different densities. When the crowd density is large, the speed changes faster, the more likely it is to produce large fluctuations in the crowd, and the more likely it is to produce crowd crowding events. In addition, the larger the area of the system occupied by high-density people, the higher the entropy value of the system will be, the more chaotic the system will be, and the higher the probability of crowd crowding events will be. All these are consistent with the actual situation. The crowd status judgment capacity of model is validated by the comparison between the results of the entropy model and the simulation.

The model can be combined with the warning system, and the warning level can be established according to the crowd status. The warning level of congested events is divided into three levels, red alert, orange alert and yellow alert. When the confusion crowed state is observed in the population, the red alert is started. The corresponding emergency evacuation measures are taken to reduce the system entropy value gradually. When the intermittent flow is observed, the orange alert will be started. Then, the crowd speed and density variation will be observed to prevent the occurrence of emergencies. When the maximum flow is observed, the yellow alert is started, and measures should be taken to prevent crowd development to the intermittent flow state. When the free flow is observed, it is considered to be normal and safe.

One of the advantages of this study is that entropy theory is used to prevent crowding events, which promotes the integration of thermodynamic entropy theory and other fields. It is well known that crowded events are prone to occur in high-density groups. So in this paper, entropy value, which can represent the event chaos degree, is used to build an early warning model of crowd crowding events. When $S > 746.4$, it means that the crowd is in a crowded state of high density. At this time, the crowded area management should be strengthened, and the crowd should be guided and diverted in time to prevent the crowd density in local areas from increasing again. Taking these into account, our study can more clearly reflect the changes of the population at each stage from low density to high density, prevent accidents and timely take emergency measures.

The model needs more simulations to compare the results fully. In the future, this model and early warning system can be further combined with the automatic control system to formulate the corresponding warning level and take more effective emergency measures according to the change of crowd status and rapid response.

## Conclusions

The crowd system is a complex system described by multiple parameters. The speed and density of people in different areas of the crowd system change instantly, which makes it difficult to predict the flow state of the crowd. This paper establishes a crowd state judgment model based on entropy theory. Entropy theory is used to combine the basic parameters of

individuals with the macro state of the population. Entropy value is used to represent the chaos degree of the population system. The state of the population can be determined by the system entropy value without recognizing and tracking the movement. It has been proved that this model can well represent the actual group situation in the state of high density and effectively avoid the occurrence of congestion events.

## Supporting information

**S1 Data.**
(XLSX)

**S2 Data.**
(XLSX)

## Acknowledgments

The authors take responsibility for the content in and drafting of the paper.

## Author Contributions

**Conceptualization:** Guomin Zhao.

**Data curation:** Guomin Zhao.

**Formal analysis:** Guomin Zhao, Cong Li.

**Funding acquisition:** Guomin Zhao, Jing Zhang.

**Investigation:** Guomin Zhao, Cong Li, Guangji Xu.

**Methodology:** Guomin Zhao, Cong Li, Guangji Xu, Falong He.

**Project administration:** Guomin Zhao, Cong Li, Guangji Xu, Falong He, Jing Zhang.

**Resources:** Guomin Zhao, Cong Li, Falong He.

**Supervision:** Guomin Zhao, Guangji Xu, Jing Zhang.

**Validation:** Cong Li.

**Visualization:** Cong Li.

**Writing – original draft:** Cong Li.

**Writing – review & editing:** Falong He, Jing Zhang.

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
