## [Decision Letter · Decision Letter 0]

23 Dec 2020

PONE-D-20-31256

A high-density crowd state judgment model based on entropy theory

PLOS ONE

Dear Dr. He,

Thank you for submitting your manuscript to PLOS ONE. After careful consideration, we feel that it has merit but does not fully meet PLOS ONE’s publication criteria as it currently stands. Therefore, we invite you to submit a revised version of the manuscript that addresses the points raised during the review process.

We look forward to receiving your revised manuscript.

Kind regards,

Mohammad Asghari Jafarabadi

Academic Editor

PLOS ONE

Reviewers' comments:

Reviewer's Responses to Questions

**Comments to the Author**

1. Is the manuscript technically sound, and do the data support the conclusions?

Reviewer #1: Partly

Reviewer #2: Partly

2. Has the statistical analysis been performed appropriately and rigorously? 

Reviewer #1: No

Reviewer #2: Yes

3. Have the authors made all data underlying the findings in their manuscript fully available?

Reviewer #1: Yes

Reviewer #2: Yes

4. Is the manuscript presented in an intelligible fashion and written in standard English?

Reviewer #1: Yes

Reviewer #2: Yes

5. Review Comments to the Author

Reviewer #1: The study “A high-density crowd state judgment model based on entropy theory” is interesting.

In this paper, a local area entropy generation model under high density state was established, and the entropy value is used to predict the crowding event in this area at a certain time in the future. From the formula of population system entropy, it can be concluded that the entropy of the system is not only related to the population density, but also related to the actual area of the system occupied by people with different densities. When the crowd density is large, the speed changes faster, the more likely it is to produce large fluctuations in the crowd, and the more likely it is to produce crowd crowding events. In addition, the larger the area of the system occupied by high-density people, the higher the entropy value of the system will be, the more chaotic the system will be, and the higher the probability of crowd crowding events will be. All these are consistent with the actual situation.

The paper is well set, and the contents are clearly described. The authors almost achieved their objectives. However, the following suggestions should be incorporated before resubmitting the paper.

1. In abstract the results should be included and discussed simultaneously.

2. The contribution of this work must be clearly describe in introduction section.

3. Figure 1 is confusing make a proper diagram or flow chart which can highlight all states and clearly describe the direction or flow of each state.

4. Page 14 of 19 correct the statement crowd disturbance .

5. Is it possible the models used in Figure 4 could describe as well.

6. Where is the Green Guide Chart is there any reference or computation table available if yes please mention. “Table 3: Data flow calculated using the Green Guide chart”

7. A new section of results discussion must be added.

Reviewer #2: I have passed through the whole content of the proposed model which is entitled:"A high-density crowd state judgment model based on entropy theory" and I see that it could be published after doing the minor corrections as follows:

1- In the abstract section, the author/s have mentioned to So>0 (line 6), is it assumption, if not from where he got it.

2- In section 2, High density crowd and state personnel, the author mentioned that both parameters involved in this model; the safety critical density of a stationary population and the safety critical density of a moving population are 4.7 and 4 persons/m^2. My suggestion is that they should have named the method or technique used in Ref. [11] to calculate these parameters.

3- In their study focused on investigating the proposed model in the industrial countries as a special case. The general formula which can be used to calculate all significant parameters involved in future outlook related to the same problem.

4- In Figure 3, caption must have more explanation by specifying each color representing the type of pressure causing crowdy crowd (from low to high). Figure's 3 caption is not clear enough for the reader.

Note: This article could be accepted and published online after doing the minor corrections.

6. PLOS authors have the option to publish the peer review history of their article (what does this mean?). If published, this will include your full peer review and any attached files.

Reviewer #1: No

Reviewer #2: No

---

## [Author Response · Author response to Decision Letter 0]

20 Jan 2021

Dear reviewers:

Thank you for your letter and for the reviewers’ comments concerning our manuscript entitled “A high-density crowd state judgment model based on entropy theory” (ID: PONE-D-20-31256). Those comments are all valuable and very helpful for revising and improving our paper, as well as the important guiding significance to our researches. We have studied comments carefully and have made correction which we hope meet with approval. The revised parts have been marked out in the paper. The main corrections in the paper and the responds to the reviewer’s comments are as following: Responds to the reviewer’s comments:

Reviewer #1:

1.Response to comment: In abstract the results should be included and discussed simultaneously.

Response: We have made correction according to the Reviewer’s comments. The research results are supplemented in the abstract. Because this paper focuses on the model, the correctness and practicability of the model are mainly described in the abstract.

2.Response to comment: The contribution of this work must be clearly describe in introduction section.

Response: As for the problems you mentioned, we have revised the introduction. The contribution of this paper is that, by further analyzing the shortcomings of previous studies, it is found that few scholars apply entropy theory to study the behavior of high-density people. The model of high-density crowd state judgment based on entropy theory established in this paper can make up for this deficiency and promote the study of entropy theory and crowd behavior.

3.Response to comment: Figure 1 is confusing make a proper diagram or flow chart which can highlight all states and clearly describe the direction or flow of each state.

Response: We are very sorry for the error in Figure 1 in the article. We have changed Figure 1 and adjusted the position of Figure 1 according to your suggestion, so we hope to get your approval.

4. Response to comment: Page 14 of 19 correct the statement crowd disturbance.

Response: We are very sorry for our incorrect writing. We have amended the term" the statement crowd disturbance" uniformly to " crowed state of confusions". It also corresponds to Figure 1.

5. Response to comment: Is it possible the models used in Figure 4 could describe as well.

Response: The model shown in Figure 4 is to analyze the relationship between population density and population density, so as to better study the change process of population status. As for your suggestion, we have carefully modified the analysis of the research results of the model in Figure 4 and hope to get your approval.

6. Response to comment: Where is the Green Guide Chart is there any reference or computation table available if yes please mention. “Table 3: Data flow calculated using the Green Guide chart”

Response: We are very sorry to have confused you. We have modified the title in Table 3. Crowd flow is calculated using the method given in reference [16-17] based on crowd flow = crowd density * crowd velocity, hence, the title of Table 3 is Data flow using the Green Guide. Data are selected from the Ando model. The Ando model is the study of population speed and density under the condition of high density of population, which is suitable for this paper.

7. Response to comment: A new section of results discussion must be added.

Response: According to your suggestion, we have supplemented the discussion at the front of the conclusion section and revised the conclusion.

Reviewer #2:

1.Response to comment: In the abstract section, the author/s have mentioned to S_0>0 (line 6), is it assumption, if not from where he got it.

Response: We are deeply sorry for our mistake. S_0 refers to entropy in the initial stat and S should refer to other stages, which we have modified. Entropy represents the degree of chaos of an event. The initial state is stable. With the development of an event, the greater the degree of chaos, the greater the entropy.

2.Response to comment: In section 2, High density crowd and state personnel, the author mentioned that both parameters involved in this model; the safety critical density of a stationary population and the safety critical density of a moving population are 4.7 and 4 persons/m^2. My suggestion is that they should have named the method or technique used in Ref. [11] to calculate these parameters.

Response: The continuous pedestrian flow model in the reference [11] you mentioned is built through the mathematical relationship between pedestrian speed and crowd density. This paper is a model to judge crowd state through entropy theory. As for the naming problem you raised, we have modified the unit of population density to pedestrians/m2 (ped/m2). Hope to get your approval.

3.Response to comment: In their study focused on investigating the proposed model in the industrial countries as a special case. The general formula which can be used to calculate all significant parameters involved in future outlook related to the same problem.

Response: Your suggestion is very important and will be a new direction for our future research.

4.Response to comment: In Figure 3, caption must have more explanation by specifying each color representing the type of pressure causing crowdy crowd (from low to high). Figure's 3 caption is not clear enough for the reader.

Response: We gratefully appreciate for your valuable suggestion. We have modified the title of Figure 3 and the graphic analysis. The pressure from the outer ring to the inner ring changes from small to large, with the minimum pressure at the edge and the highest pressure at the center.

I tried our best to improve the manuscript and made some changes in the manuscript. I appreciate for Editors warm work earnestly, and hope that the correction will meet with approval.

 Once again, thank you very much for your comments and suggestion.

 Best regards,

 Falong He

---

## [Decision Letter · Decision Letter 1]

5 Mar 2021

PONE-D-20-31256R1

A high-density crowd state judgment model based on entropy theory

PLOS ONE

Dear Dr. He,

Thank you for submitting your manuscript to PLOS ONE. After careful consideration, we feel that it has merit but does not fully meet PLOS ONE’s publication criteria as it currently stands. Therefore, we invite you to submit a revised version of the manuscript that addresses the points raised during the review process.

Please address the concerns of reviewer #3.

We look forward to receiving your revised manuscript.

Kind regards,

George Vousden

Senior Editor

PLOS ONE

On behalf of,

Mohammad Asghari Jafarabadi

Academic Editor

PLOS ONE

Journal Requirements:

Reviewers' comments:

Reviewer's Responses to Questions

**Comments to the Author**

1. If the authors have adequately addressed your comments raised in a previous round of review and you feel that this manuscript is now acceptable for publication, you may indicate that here to bypass the “Comments to the Author” section, enter your conflict of interest statement in the “Confidential to Editor” section, and submit your "Accept" recommendation.

Reviewer #1: All comments have been addressed

Reviewer #3: (No Response)

2. Is the manuscript technically sound, and do the data support the conclusions?

Reviewer #1: Yes

Reviewer #3: Partly

3. Has the statistical analysis been performed appropriately and rigorously? 

Reviewer #1: Yes

Reviewer #3: N/A

4. Have the authors made all data underlying the findings in their manuscript fully available?

Reviewer #1: Yes

Reviewer #3: Yes

5. Is the manuscript presented in an intelligible fashion and written in standard English?

Reviewer #1: Yes

Reviewer #3: No

6. Review Comments to the Author

Reviewer #1: All of my comments from the previous review round have been met. Now i am suggesting the acceptance of the paper in the current form.

Reviewer #3: The topic of this paper seems interesting, and the authors have revised their manuscript according to the former reviewers. However, according to my reading, I think there are still some problems:

1) The written style of this paper should be revised to make it widely readerable.

2) The authors established a high-density crowd state judgment model based on entropy theory, I suggest the authors to give more details on how to applied their model to deal real-world problems.

3) some sentences are unreaderable, the authors should carefully revised their paper. For example : "Negative

entropy flow obtained by a system from the outside world can reduce the total entropy, so that the system

to the orderly direction of development, the system to maintain a stable state."

7. PLOS authors have the option to publish the peer review history of their article (what does this mean?). If published, this will include your full peer review and any attached files.

Reviewer #1: No

Reviewer #3: No

---

## [Author Response · Author response to Decision Letter 1]

9 Apr 2021

Dear reviewers:

Thank you for your letter and for the reviewers’ comments concerning our manuscript entitled “A high-density crowd state judgment model based on entropy theory” (ID: PONE-D-20-31256). Those comments are all valuable and very helpful for revising and improving our paper, as well as the important guiding significance to our researches. We have studied comments carefully and have made correction which we hope meet with approval. The revised parts have been marked out in the paper. The main corrections in the paper and the responds to the reviewer’s comments are as following: Responds to the reviewer’s comments:

Reviewer #3:

1.Response to comment: The written style of this paper should be revised to make it widely readerable.

Response: We have made correction according to the Reviewer’s comments. We have modified it according to the comments of reviewers. Some modifications have been made in the format and language of the paper, which we hope will make the paper more readable.

2.Response to comment: The authors established a high-density crowd state judgment model based on entropy theory, I suggest the authors to give more details on how to applied their model to deal real-world problems. 

Response: As for the application of the model you mentioned, we have mainly supplemented it in the chapter of case analysis. In real-world problems, the model can be combined with the warning system to prevent the occurrence of crowd crowding events. According to the change of crowd status, the warning level will be divided into three levels: red alert, orange alert and yellow alert. Different levels of early warning take different emergency rescue measures to ensure the safety of the people.

3.Response to comment: some sentences are unreaderable, the authors should carefully revised their paper. For example: "Negative entropy flow obtained by a system from the outside world can reduce the total entropy, so that the system to the orderly direction of development, the system to maintain a stable state."

Response: We are very sorry that some of the sentences in the article are difficult for you to understand. We have revised the example sentences and some of the language in the article, hoping to make the article more readable and more accurate.

We tried our best to improve the manuscript and made some changes in the manuscript. We appreciate for Editors warm work earnestly, and hope that the correction will meet with approval.

 Once again, thank you very much for your comments and suggestion.

 Best regards,

 Falong He

---

## [Decision Letter · Decision Letter 2]

19 May 2021

PONE-D-20-31256R2

A high-density crowd state judgment model based on entropy theory

PLOS ONE

Dear Dr. He,

Thank you for submitting your manuscript to PLOS ONE. After careful consideration, we feel that it has merit but does not fully meet PLOS ONE’s publication criteria as it currently stands. Therefore, we invite you to submit a revised version of the manuscript that addresses the points raised during the review process.

We look forward to receiving your revised manuscript.

Kind regards,

Mohammad Asghari Jafarabadi

Academic Editor

PLOS ONE

Reviewers' comments:

Reviewer's Responses to Questions

**Comments to the Author**

1. If the authors have adequately addressed your comments raised in a previous round of review and you feel that this manuscript is now acceptable for publication, you may indicate that here to bypass the “Comments to the Author” section, enter your conflict of interest statement in the “Confidential to Editor” section, and submit your "Accept" recommendation.

Reviewer #1: All comments have been addressed

Reviewer #4: All comments have been addressed

2. Is the manuscript technically sound, and do the data support the conclusions?

Reviewer #1: Yes

Reviewer #4: No

3. Has the statistical analysis been performed appropriately and rigorously? 

Reviewer #1: Yes

Reviewer #4: No

4. Have the authors made all data underlying the findings in their manuscript fully available?

Reviewer #1: Yes

Reviewer #4: No

5. Is the manuscript presented in an intelligible fashion and written in standard English?

Reviewer #1: Yes

Reviewer #4: No

6. Review Comments to the Author

Reviewer #1: The authors tried our best to improve the manuscript and made some changes in the manuscript. I accept the paper for publication in the previous round of review. While still i am suggesting to accept the paper for publication.

Reviewer #4: -The paper is very hard to read. There is no clear sense of progress in the reading. Some paragraphs are confusing

-The biggest caveat it that it does not distinguish between what is already know in literature and want are the new results produced by the authors. Is it that the literature focuses mainly on density <= 4 ped/m2 and that the authors focus on higher density? If so, it should be stated clearly.

- All the formulas in the text: where do they come from? how were they derived?

- How were the numbers in tables and in text (ex: in section "Case Analysis") derived?

- p. 21: what is the unit for S?

-What methods were used in the study?

-The authors claim for instance that "All these results are consistent with the actual situation (p. 21) ", "The results show that the models are effective" (Abstract); but nowhere is there a proof of such a claim. In other words: was the model validated?

-The list of the 4 types (free flow, maximum flow, intermittent flow, crowded state of confusions) is shown at 2 different places (p. 9 and p. 16)

- There is a lot of typos on words (ex: "willimpact" instead of "will impact"), on formulas (ex: S?S2 instead of S>S2).

7. PLOS authors have the option to publish the peer review history of their article (what does this mean?). If published, this will include your full peer review and any attached files.

Reviewer #1: No

Reviewer #4: No

---

## [Author Response · Author response to Decision Letter 2]

2 Jul 2021

Dear reviewers:

Thank you for your letter and for the reviewers’ comments concerning our manuscript entitled “A high-density crowd state judgment model based on entropy theory” (ID: PONE-D-20-31256). Those comments are all valuable and very helpful for revising and improving our paper, as well as the important guiding significance to our researches. We have studied comments carefully and have made correction which we hope meet with approval. The revised parts have been marked out in the paper. The main corrections in the paper and the responds to the reviewer’s comments are as following: Responds to the reviewer’s comments:

Reviewer #1:

1.Response to comment: The authors tried our best to improve the manuscript and made some changes in the manuscript. I accept the paper for publication in the previous round of review. While still i am suggesting to accept the paper for publication.

Response: Thank you very much for accepting my article.

Reviewer #4:

1.Response to comment: The biggest caveat it that it does not distinguish between what is already know in literature and want are the new results produced by the authors. Is it that the literature focuses mainly on density <= 4 ped/m2 and that the authors focus on higher density? If so, it should be stated clearly. 

Response: 4p/m2 is the safety critical density of the movement group. This paper studies the prediction model of the crowded events in high density state with density greater than 4p/m2. Therefore, by analyzing and summarizing the previous studies of scholars and combining with the change of pedestrian speed, the state of pedestrians is divided to achieve the purpose of preventing dangerous events.

2.Response to comment: All the formulas in the text: where do they come from? how were they derived?

Response: Some well-known entropy formulas, no references. The rest is basically the author's mathematical derivation, for such formulas, the author has a detailed written expression.

3.Response to comment: How were the numbers in tables and in text (ex: in section "Case Analysis") derived?

Response: The numbers in Table 3 are calculated according to the Green Guide research conclusions and the Ando model. Table 4 is based on Fig 3 and Table 3. In the "Case Study" section, the numbers are calculated by assuming the radius of the study area and by combining the mathematical formulas in the model with the range of population conditions. The remaining numbers in this paper are mainly obtained by summarizing and analyzing the references noted. 

4.Response to comment: p. 21: what is the unit for S?

Response: Entropy in this paper is non-thermodynamic entropy, which is a value representing the disorder degree of an object without a clear unit.

5.Response to comment: What methods were used in the study?

Response: It has been clearly stated in the abstract that this paper adopts the thermodynamic entropy theory and the local equilibrium hypothesis. By analyzing and summarizing the previous studies on velocity and density as well as mathematical derivation, the entropy judgment model is established. The research methods are statistical analysis and mathematical derivation.

6.Response to comment: The authors claim for instance that "All these results are consistent with the actual situation (p. 21) ", "The results show that the models are effective" (Abstract); but nowhere is there a proof of such a claim. In other words: was the model validated?

Response: In response to your suggestion, we have added a chapter of “Model validation”. The simulation results show that there is a relationship between the crowd density, the crowd area in different conditions and the crowd system entropy. In addition, in the “Case analysis”, the calculation process of the model is clearly expressed for the convenience of readers. The results of case analysis and model verification are the same, which further proves the reliability and effectiveness of the model.

7.Response to comment: The list of the 4 types (free flow, maximum flow, intermittent flow, crowded state of confusions) is shown at 2 different places (p. 9 and p. 16)

Response: The four types of lists (free flow, maximum flow, intermittent flow, crowded state of confusions) displayed in two different places have different meanings. The former explains the four crowd states mainly studied in this paper, while the latter explains the speed, density and entropy corresponding to the four crowd states in detail.

8.Response to comment: There is a lot of typos on words (ex: "willimpact" instead of "will impact"), on formulas (ex: S?S2 instead of S>S2).

Response: We are very sorry that some of the sentences in the article are difficult for you to understand. We have revised the example sentences and some of the language in the article, hoping to make the article more readable and more accurate.

We tried our best to improve the manuscript and made some changes in the manuscript. We appreciate for Editors warm work earnestly, and hope that the correction will meet with approval.

 Once again, thank you very much for your comments and suggestion.

 Best regards,

 Falong He

---

## [Decision Letter · Decision Letter 3]

19 Jul 2021

A high-density crowd state judgment model based on entropy theory

PONE-D-20-31256R3

Dear Dr. He,

We’re pleased to inform you that your manuscript has been judged scientifically suitable for publication and will be formally accepted for publication once it meets all outstanding technical requirements.

Kind regards,

Mohammad Asghari Jafarabadi

Academic Editor

PLOS ONE

Reviewers' comments:

Reviewer's Responses to Questions

**Comments to the Author**

1. If the authors have adequately addressed your comments raised in a previous round of review and you feel that this manuscript is now acceptable for publication, you may indicate that here to bypass the “Comments to the Author” section, enter your conflict of interest statement in the “Confidential to Editor” section, and submit your "Accept" recommendation.

Reviewer #4: All comments have been addressed

Reviewer #5: All comments have been addressed

2. Is the manuscript technically sound, and do the data support the conclusions?

Reviewer #4: Yes

Reviewer #5: Yes

3. Has the statistical analysis been performed appropriately and rigorously? 

Reviewer #4: Yes

Reviewer #5: Yes

4. Have the authors made all data underlying the findings in their manuscript fully available?

Reviewer #4: Yes

Reviewer #5: Yes

5. Is the manuscript presented in an intelligible fashion and written in standard English?

Reviewer #4: Yes

Reviewer #5: Yes

6. Review Comments to the Author

Reviewer #4: (No Response)

Reviewer #5: All comments of reviewers have been addressed correctly one by one. I believe that manuscrip is improved now.

7. PLOS authors have the option to publish the peer review history of their article (what does this mean?). If published, this will include your full peer review and any attached files.

Reviewer #4: No

Reviewer #5: **Yes: **Masoud Amiri

---

## [Editor Report · Acceptance letter]

24 Aug 2021

PONE-D-20-31256R3 

A high-density crowd state judgment model based on entropy theory 

Dear Dr. He:

I'm pleased to inform you that your manuscript has been deemed suitable for publication in PLOS ONE. Congratulations! Your manuscript is now with our production department. 

Kind regards, 

on behalf of

Professor Mohammad Asghari Jafarabadi 

Academic Editor

PLOS ONE